# Investigation of the JPA-Bandwidth Improvement in the Performance of the QTMS Radar

**DOI:** 10.3390/e25101368

**Published:** 2023-09-22

**Authors:** Milad Norouzi, Jamileh Seyed-Yazdi, Seyed Mohammad Hosseiny, Patrizia Livreri

**Affiliations:** 1Department of Physics, Faculty of Science, Vali-e-Asr University of Rafsanjan, Rafsanjan 7718897111, Iran; miladnorouzi.edu@gmail.com; 2Department of Physics, Faculty of Science, University of Urmia, Urmia 5756151818, Iran; 3Department of Engineering, University of Palermo, Viale delle Scienze Bldg. 9, 90128 Palermo, Italy; patrizia.livreri@unipa.it

**Keywords:** engineered JPA, quantum illumination, quantum correlation, QTMS radar, SNR

## Abstract

Josephson parametric amplifier (JPA) engineering is a significant component in the quantum two-mode squeezed radar (QTMS) to enhance, for instance, radar performance and the detection range or bandwidth. We simulated a proposal of using engineered JPA (EJPA) to enhance the performance of a QTMS radar. We defined the signal-to-noise ratio (SNR) and detection range equations of the QTMS radar. The engineered JPA led to a remarkable improvement in the quantum radar performance, i.e., a large enhancement in SNR of about 6 dB more than the conventional QTMS radar (with respect to the latest version of the QTMS radar and not to the classical radar), a substantial improvement in the probability of detection through far fewer channels. The important point in this work was that we expressed the importance of choosing suitable detectors for the QTMS radars. Finally, we simulated the transmission of the signal to the target in the QTMS radar and obtained a huge increase in the QTMS radar range, up to 482 m in the current study.

## 1. Introduction

In general, radars transmit radio waves to one or more targets using transmitter antennas and receive and measure echoes using receiver antennas to detect the presence or absence of targets using a detector. Many factors, such as noise and clutter, can be mentioned that call into question this simplicity [1]. The difference between a quantum radar (QR) and a classical radar (CR) can be deduced even from their names. Basic quantum concepts such as the correlation between a pair of entangled signals are present in QRs [1,2,3,4,5,6,7,8,9,10,11,12,13,14,15,16,17,18,19]. The discussion about QRs has flourished for several years. Several research teams, including Balaji et al. [1,2,3] and Barzanjeh et al. [15], etc., have implemented operational prototypes of QR. All the results obtained by these teams showed a significant improvement in the performance of QR, compared to the CR counterpart [2,3,4,5,6,7,8,9,15,19]. A 6 dB improvement in transmission power [11], 4 to 6 dB enhancement in SNR [2,3,4,5,6,7,8,9,15,19,20], and an improvement of 6 dB in the QR receiver operating characteristic (ROC) curve [19,20], compared to CR, can be considered. The quantum illumination (QI) range, which gives the accuracy of the square of the mean of the delay of range, can be tens of dB above that of a CR counterpart with the same bandwidth and transmitted energy [20]. In the QR, the samples are much lower than in the CR, and the signal in QR has a higher correlation coefficient than its classical counterpart [2,3,4,5,6,7,8,9,15,19]. The QRs can also be made impervious to hackers by using quantum cryptography [9]. This means that it is possible to create a secure channel by encoding the transmitted photons to protect the information against eavesdropping [9]. The target may also be more visible by using QR rather than CR, due to a quantum effect on the radar cross-section [21]. One type of QR is the quantum two-mode squeezed (QTMS) radar, which is very similar to conventional noise radars [1,2,3]. These radars use the Josephson parametric amplifier (JPA) and can produce the signal and idler directly in the microwave band [1,2,3,6,8,11,12,15,16]. On the other hand, one of the current disadvantages of QTMS radars is their very high costs of implementation and equipment [1,2,3,6,8,11,12,15,16]. In recent articles [1,2,3,6,8,11,12,15], it was observed that JPAs have limitations such as low bandwidth, and therefore, we need to engineer them to improve the performance of the QR. Hence, engineering JPAs in QTMS radars gives us the capabilities to build high-range QTMS radars. One of the most important issues for engineers is the range measurement of a QR. Therefore, in this study, the range equation of a QTMS radar is introduced, and the results are reviewed. We simulate a QTMS radar proposal with a larger bandwidth, better detection range, and improved SNR, taking into account the prototype of the QTMS radar implemented in [1,3] and using the EJPA [22]. The QTMS radar has shown much more enhancement from the classical than the 6 dB [2,3,4,5,6,7,8,9,15,19,20], and this paper purports to have an additional 6 dB in SNR from those results. In this work, we present a simulation study of a QR inspired by quantum illumination, which requires only independent measurement of the signal and idler. After introducing QR and the basic principles of the QTMS radars, we present and evaluate the EJPA and use it to simulate the radar’s design. Finally, after post-processing, the results are presented to show and confirm the capability of our design.

## 2. Preliminaries

### 2.1. Quantum Radar (QR)

The basis of the work of a QR can be summarized as follows [1,2,3,4,5,6,7,8,9,10,11,12,13,14,15,16,17,18,19]:Using a pump and a signal generator, we produce a current of entangled photon pairs (signal/idler) using quantum sources.To send a signal to the target, we need to amplify the signal with low-noise amplifiers, and to determine the presence of the target, we have to record the idler.After receiving the signal reflected from the target by the receiver antenna, the signal and idler are amplified again, and an analog-to-digital conversion (ADC) is applied.Using a suitable detector, the presence or absence of a target can be inferred. Figure 1 depicts the general block diagram of a quantum radar.

#### 2.1.1. QTMS Radar

A type of QR that we considered in the template is the quantum two-mode squeezed (QTMS) radar, which is used as the operational prototype that was introduced in [1,2,3,5,6,8,15]. Here, the term squeezed refers to the electromagnetic field state that decreases the uncertainty of one component of the field relative to the coherent state (uncertainty in the amplitude and phase of the electric field are the same), increasing uncertainty in the other component [1,3,23]. In other words, the quantum noise decreases in linear compounds of some of the quadratures and increases in other compounds, and squeezing appears [1,2,3,5,6,8]. In this paper, we deal with the correlating and squeezing in-phase (I) and quadrature (Q) voltages. We can brief the operation of the QTMS radar as follows:Utilize the JPA to generate a pair of entangled signals (signal and idler). Amplify both the signal and the idler. Transmit the signal through the free space forward to the target. Perform a heterodyne measurement on the idler, and hold a record of the results in the form of a time series of I and Q voltages.Receive a reflected signal. Fulfill a heterodyne measurement on it to create a time series of I and Q voltages.Correlate the I and Q voltages of the signal and idler.If the correlation surpasses a preset threshold, notify a detection.

The main part of QTMS radars is the source of entanglement generation, the Josephson parametric amplifier (JPA). JPAs are devices that generate a two-mode squeezed vacuum (TMSV) state [1,2,3,5,6,8]. JPAs are placed in dilution refrigerators for two reasons: first, because they have a resonant cavity with a superconducting quantum interferometer device (SQUID) and superconducting properties and second, to prevent noise absorption in the entangled signal [1,2,3,5,6,8,15]. Figure 2 shows a schematic representation of the JPA.

### 2.2. Two-Photon Entanglement

A covariance matrix is a matrix whose elements show a correlation between different system parameters. A special type of correlation is entanglement [24,25,26]. When two beams of light are entangled, they have very strong correlations. The correlation power resulting from quantum entanglement simply cannot occur in classical physics [1,3]. The root of entanglement is in the quantum superposition principle and has no classical counterpart [26]. In general, in a quantum state, if the measurement of the first qubit affects the result of the measurement of the second qubit, we have an entangled state. Otherwise, it is non-entangled [24,26].

As mentioned in Section 2.1.1, since the photons of the signal and idler originate from the same pump photon, there is a strong quantum correlation between the signal and idler, resulting in the squeezing of I and Q voltages [1,2,3,5,6,8,15]. It is important to emphasize that squeezing is not solely a consequence of entanglement.

For better detection and measurement with classical instruments, the signal and idler need to be amplified in stages. Unfortunately, amplification adds a lot of noise and weakens the entanglement. Entanglement can easily be eliminated by factors such as loss (e.g., antenna gain) and noise (e.g., the presence of amplifiers) [13,27,28,29]. However, as shown in experimental results [1], there is a quantum enhancement (in the form of higher correlations) in the QTMS radar. In QTMS radars, we utilized the entanglement of continuous variables of light squeezed by the JPA entanglement generation source [1,2,3,30].

## 3. Results and Discussion

### 3.1. Engineering JPA (EJPA)

The JPAs are commonly used as narrowband signal amplifiers, meaning that they have limitations (i.e., narrow bandwidth) that prevent the improvement of their performances [1,2,3,5,6,8,15,22]. Our EJPA is similar to that described in [22], where a broadband EJPA by the pumped flux impedance method was presented. Therefore, we used it to simulate the quantum radar. One of the advantages of this JPA is the wide bandwidth at low gain rates [22]. By matching the impedance with the input amplifier, its bandwidth is significantly increased from 1 MHz to 300 MHz. The input signal is reflected as an amplified output signal, with a gain of about 20 dB [22]. Figure 3 shows a schematic representation of the equivalent circuit of an EJPA device, in which the entire device is fabricated integrally on intrinsic silicon (>10 kΩcm resistivity). The device operates in a dilution refrigerator with a base temperature of 7 mK [22]. A SQUID loop is made with two Josephson junctions placed in parallel on each side. If the flux line on the chip is combined with two Josephson junctions, flux pumping is provided. The λ/4 resonator with a characteristic impedance of 45 Ω reduces the JPA resonant quality factor. On the other hand, the λ/2 resonator with a characteristic impedance of 80 Ω reduces the frequency dependence in the system sensitivity matrix, and this leads to the amplification of the bandwidth [22].

A parallel plate capacitor with two top and bottom electrodes with a total input capacity c = 2.03 ± 0.02 pF is located at the JPA input, which is connected to the ground. The input line is galvanically connected to the lower electrode by impedance, which is directly connected to the SQUID. The upper electrode of the capacitor is connected to the ground in parallel with the SQUID with a non-galvanic connection. By connecting galvanically to the input, the coupling quality coefficient decreases, and therefore, the amplification bandwidth increases [22]. The JPA assumed in the current study is the degenerate four-wave type, which means that the input and output frequencies are identical. The signal (and idler) frequency here is 5.31 GHz [22].

### 3.2. A Proposed QTMS Radar Design with EJPA

The block diagram of the QTMS radar is illustrated in Figure 4. The JPA bandwidth is increased from 1 MHz [1,3] by the pumped flux impedance method to 300 MHz. The pump power is 5 dBm. The JPA is located inside the refrigerator, as shown in Figure 4, and is connected to the T-bias via a microwave switch (which can turn the pump on or off) and a device called a shot-noise tunnel junction (SNTJ). These are considered a part of the calibration process to confirm the entanglement of the JPA output signal [1,2,3,5,6,8]. The output signal is amplified to make it easier to measure and detect. Because amplifiers add noise to the signal, and usually, the added noise is from the first amplifier, a high-electron-mobility transistor (HEMT) (which is a low-noise amplifier) and a semiconductor amplifier [1,2,3,5,6,8,15] should be used. HEMT is also placed in a dilution refrigerator, and semiconductor amplifiers operate at 4 K. Our suggestion to experimental researchers regarding the dilution refrigerator is the Bluefors LH250, which uses liquid helium for cooling. It is used in experiments in which the cooling power of this refrigerator for a temperature of 7 mK is about 10 μW [31]. 

After calibration, circulators are placed, which act as insulators for our system to prevent additional signal and noise from reaching the JPA [1,2,3,5,6,8,15]. After amplification, the signal is sent to the target by the C-band antenna, but we record the idler. It should be noted that no measurements are performed on the idler before the signal arrives. The local oscillators (LO), ${\mathrm{LO}}_{1}$, and ${\mathrm{LO}}_{2}$ are applied to the reflection signal of the target and the idler to convert the frequency to the intermediate frequency (20 MHz) [15]. Finally, the signal is amplified again and detected after digitization. 

Note that in this work, we do not have a delay line, but there is a time delay to measure the signal and idler [1]. The two pulses are measured at different times, with respect to maintaining the correlation between them. The time delay between the two pulses is due to the length of the free-space path of the transmitted signal [1,32,33].

The main idea of using EJPA comes from the fact that we need a high-range QR in practice. Our EJPA has three wings that are very useful for improving our simulated quantum radar: first, high bandwidth; second, high power; and third, low gain. In the design of a QR, special attention should be paid to very important points, including various parameters of the radar. For example, to implement a QR with a long-range, we need high signal power. Of course, we need to know how high the power should be so as not to suppress the correlation. What antennas should we use with what gains, or which amplifiers should we use so that less noise will be introduced into the system? Therefore, all the parameters of a QR must match each other to find an improvement in its performance. In this paper, we investigate the improvement of a QR according to the sensitivity of choosing different parameters.

### 3.3. Post-Processing

As shown in Figure 5A,B, the idler and signal modes are recorded after amplification with an analog-to-digital converter (ADC) card. (Our suggestion to experimental researchers is to use a dual-channel ADC AD570JD with 8-bit resolution) [15]. The recorded data from the ADC is split into shorter arrays of copies. To derive the measurement statistics of the signal and idler mode quadratures, the digital fast Fourier transform (FFT) at idler ($\omega_{l}$) and signal frequencies ($\omega_{s}$) after analog down conversion can be used on each array separately. These measurement results are useful to compute the covariances of the signal and idler modes. 

The M copies of the signal and idler modes from ADC are sent to the receiver (Figure 5B). The task of the beam splitter (50:50) is to mix the signal mode reflected from the target with the locally detected idler mode. The outputs of the beam splitter are detected (which are used as the input to a threshold detector). (Output is the target absence or presence decision.) According to Figure 5, ${\hat{U}}_{\eta ,S}^{\left(k\right)}$ (received signal mode 1≤k≤M) and ${\hat{U}}_{I}^{\left(k\right)}$ (received idler mode 1≤k≤M) are mixed by a 50:50 beam splitter, and the output is as follows [15,34]:(1)$${\hat{U}}_{\eta ,\pm }^{\left(k\right)}=\frac{{\hat{U}}_{\eta ,S}^{\left(k\right)}\pm {\hat{U}}_{I}^{\left(k\right)}}{\sqrt{2}}$$

After detecting these modes, the photon counts are equal to the quantum measurements of the corresponding number operator [15,34]:(2)$${\hat{N}}_{\eta ,\pm }^{\left(k\right)}={\hat{U}}^{†}{}_{\eta ,\pm }^{\left(k\right)}{\hat{U}}_{\eta ,\pm }^{\left(k\right)}$$

The total photon counts for the two detectors are equal to
(3)$${\hat{N}}_{\eta }^{\;}=\overset{M}{\underset{k=1}{\sum }}\left({\hat{N}}_{\eta ,+}^{\left(k\right)}-{\hat{N}}_{\eta ,-}^{\left(k\right)}\right)$$

According to recent literature [15,34], the microwave mode of the signal return to post-processing (η=0 in the absence of the target, and η≠0 in the presence of the target) is equal to [15,34]:(4)$${\hat{U}}_{\eta ,S}^{\left(k\right)}=\sqrt{G_{s}}[\sqrt{\eta }{\hat{a}}_{S}^{†}+\sqrt{\frac{\eta \left(G_{s}^{A}-1\right)}{G_{s}^{A}}}{\hat{a}}_{n,S}^{A}+\sqrt{\frac{\left(1-\eta \right)}{G_{s}^{A}}}{\hat{a}}_{n}^{†E}+\sqrt{\frac{\left(G_{s}^{D}-1\right)}{G_{s}}}{\hat{a}}_{n,S}^{D}]+\sqrt{2}{\hat{a}}_{V}$$

Additionally, the idler’s microwave mode toward the post-processor is
(5)$${\hat{U}}_{I}^{\left(k\right)}=\sqrt{G_{I}}[{\hat{a}}_{I}+\sqrt{\frac{\left(G_{I}^{A}-1\right)}{G_{I}^{A}}}{\hat{a}}_{n,I}^{†A}+\sqrt{\frac{\left(G_{I}^{D}-1\right)}{G_{I}}}{\hat{a}}_{n,I}^{†D}]$$

Here, $G_{s}=G_{s}^{D}\left(dB\right)G_{s}^{A}\left(dB\right)$ is the system signal gain, where $G_{s}^{D}$ and $G_{s}^{A}$ are the detection signal gain and amplification signal gain, respectively. Additionally, $G_{I}=G_{I}^{D}\left(dB\right)G_{I}^{A}\left(dB\right)$ is the system idler gain. The effective dissipation range is −25 dB<η<0 dB. ${\hat{a}}_{S}$ and ${\hat{a}}_{I}$ are the signal and idler annihilation operators, respectively. ${\hat{a}}_{n,S}^{†A}$ and ${\hat{a}}_{n,I}^{†A}$ are the signal and idler amplification noise-creation operators at 4 K, respectively. ${\hat{a}}_{n}^{E}$ is the environment noise mode operator at room temperature, and ${\hat{a}}_{V}$ is the vacuum mode. ${\hat{a}}_{n,S}^{†D}$ and ${\hat{a}}_{n,I}^{†D}$ are the signal and idler detection creation operators at 290 K, respectively [15,34].

The detection probability in terms of the false-alarm probability for the QTMS radar with respect to detector A, which corresponds to detector 5 in [1], according to previous publications [1,2,3,5,6,8,32,33], is expressed by:(6)$$P_{DA}^{QR}=Q_{1}\left(\frac{\rho \sqrt{2N}}{1-\rho^{2}},\frac{\sqrt{-2lnP_{FA}}}{1-\rho^{2}}\right)$$
where N is the number of channels (or the number of samples integrated), $Q_{1}$ is the Marcum function [34], and the subscript *DA* denotes detector A. Furthermore, ρ is the quantum correlation coefficient and is given by [35,36,37,38,39,40,41,42,43,44,45]:(7)$$\rho =\frac{\rho_{0}}{\sqrt{1+{\left(1/SNR^{QR}\right)}^{4}}}$$
where the $SNR^{QR}$ is the signal-to-noise ratio of the QTMS radar and is defined as [15,34,36,37]:(8)$${\left(SNR\right)}_{M}^{QR}=\frac{4M\;{\left[\left({\langle {\hat{N}}_{\eta ,+}\rangle }_{H_{1}}-{\langle {\hat{N}}_{\eta ,-}\rangle }_{H_{1}}-\left({\langle {\hat{N}}_{\eta ,+}\rangle }_{H_{0}}-{\langle {\hat{N}}_{\eta ,-}\rangle }_{H_{0}}\right)\right)\right]}^{2}}{{\left(\sqrt{{\langle \Delta {\hat{N}}_{\eta }^{2}\rangle }_{H_{0}}}+\sqrt{{\langle \Delta {\hat{N}}_{\eta }^{2}\rangle }_{H_{1}}}\right)}^{2}}$$

Here, M=B·τ is the number of modes, B is the bandwidth, and τ is the integration time. Additionally, the value expected indicates the average over the total M copy. The terms in the above equation are expressed in Appendix A.

#### 3.3.1. SNR and ROC

Using Equations (8) and (A1)–(A8) (in the Appendix A)) and the corresponding parameter in Table 1, Figure 6 shows the SNR versus signal and the idler photon numbers N_s_ plot, comparing the three scenarios: conventional JPA (the latest version of the QTMS radar) [1,3], Josephson ring modulators (JRM) [15], and EJPA. The clear conclusion that can be deduced from this plot is that the SNR performs better in the EJPA scenario than in the other scenarios. The SNR for EJPA is about 5 dB better than JRM and about 6 dB better than conventional JPA. Therefore, the EJPAs are promising to improve SNR, meaning a better performance of quantum radars. It means that we have a 6 dB improvement in SNR, in addition to an improvement of 4 to 6 dB (latest version of the QTMS radar [1]) in SNR in previous works [2,3,4,5,6,7,8,9,15,19,20].

Using Equation (6), the ROC diagram for different scenarios is depicted in Figure 7 and Figure 8, which clearly show the superiority of the EJPA compared to other scenarios. In Figure 7, the probability of detection in EJPA is better than in other scenarios.

The ROC comparison plot for conventional JPA, JRM, and EJPA is illustrated in Figure 8. It is clear that the detection probability in an EJPA with a smaller number of *N* channels reaches a maximum of one, demonstrating a significant improvement.

In various articles [1,2,15,33,35], the superiority of the QTMS radars over conventional classical radars has been discussed. In this work, we only investigate the superiority of EJPAs with respect to conventional JPAs. However, in order to confirm the results according to the $P_{D}^{CR}=Q_{1}\left(\sqrt{2N(SNR)},\sqrt{-2lnP_{FA}}\right)$ equation, as discussed in [35,36], and Equation (6) for a conventional classical radar, the ROC comparison between a classical radar and QTMS with EJPA with respect to detector A is plotted in Figure 9A. Moreover, with respect to detector B, which corresponds to detector 1 in [36] with equation $P_{DB}^{QR}=(1/2)erfc\left((erfc^{-1}(2P_{FA})-\sqrt{N}\rho )/\sqrt{1+\rho^{2}}\right)$, the comparison between the QTMS radar and the conventional classical radar is depicted in Figure 9B. In Figure 9A, we see that when detector A is used, the quantum radar is not superior to the classical radar, while in Figure 9B, we see that, considering detector B, our quantum radar performs better than the classical radar. Therefore, in the quantum radar, the choice of the suitable detector is very important. Moreover, the same results are obtained for different *N* when comparing the two cases.

In recent articles [46,47], important factors in improving the performance of the QTMS radar have been investigated, with respect to the effects of decoherence and entanglement, respectively. Here, the effects of the bandwidth and quantum correlation changes on the SNR are investigated. Using Equations (8) and (A1)–(A8) in Appendix A and the corresponding parameter in Table 1, the SNR plot is also examined versus a correlation function $\langle {\hat{a}}_{S}{\hat{a}}_{I}\rangle$ in Figure 10A. This figure shows that when the correlation increases, the SNR of the QTMS radar also increases. Therefore, the correlation and entanglement play the most important roles in QTMS radars, and we need to fabricate more correlated signals and idlers (by engineering the JPAs) [38,39,40,41,42,43,44,45,48]. Furthermore, in Figure 10B, we plotted the variation of SNR in terms of correlation $\langle {\hat{a}}_{S}{\hat{a}}_{I}\rangle$ and bandwidth B. This dependence between correlation, bandwidth, and SNR is seen in Equations (A3) and (A4) in the appendix and the general Equation (8). This figure shows that JPA engineering is not only related to bandwidth; other important factors, such as maintaining correlation, are also very important in radar SNR. The key point is that improving the SNR of a QTMS radar requires high bandwidth and high correlation. Therefore, the results obtained from this figure can be very useful in the design of operational radars. This means that in addition to JPA engineering in terms of bandwidth, we need to perform a series of tasks to prevent correlation suppression, such as using low noise amplifiers, choosing the correct antennas according to their gain, and making JPAs that are acceptable according to the gain and correlation, etc.

#### 3.3.2. QTMS Radar Range

As mentioned earlier, range evaluation is one of the most important tasks of the quantum radar. Since QTMS radars are very similar to conventional noise radars, the QTMS range equation can be obtained as follows [35,49]:(9)$$R_{max}={\left(\frac{GA_{e}\sigma P_{s}}{{\left(4\pi \right)}^{2}P_{n}{\left(SNR\right)}_{min}^{Q}}\right)}^{1/4}.$$
where G is the antenna gain, λ is the wavelength, $A_{e}=G\lambda^{2}/4\pi$ is the effective antenna area, σ is the target radar cross-section, and $P_{s}$ and $P_{n}$ are the signal and noise powers, respectively. The only difference between this equation and the conventional noise radar equation is ${\left(SNR\right)}_{min}^{Q}$, which is the SNR of the QTMS radar here. Figure 11 depicts the detection range versus ${\left(SNR\right)}_{min}^{Q}$. In previous papers [1,2,3,15], the authors either transmit the signal to the target [1,2,3] or used a low-power short-range radar [15] for various applications. Instead, in this work, we can see that the radar range has increased up to 482 m. Additionally, SNR loses performance efficiency with increasing range. 

Note that a range of 482 m is just an estimation based on a large number of estimated parameters, putting together devices and apparatuses of different kinds, which means the range can reach the order of a few hundred meters.

### 3.4. Simulation Methodology

In this study, we assumed that the antenna works in the C-band (4–8 GHz). These antennas have fewer losses than X-band antennas (8–12 GHz) [1,2,3]. System parameters are calculated in Table 1 to compare with the corresponding parameters obtained from the information reported in [1,15], as follows:

The question raised here is how the parameters in Table 1 are extracted and how the simulation is performed in this paper. It can be answered by considering Equation (8), for example. All the required expressions are given in Appendix A by Equations (A1)–(A8) so that by placing them in Equation (8), we obtain a general relation for SNR, which includes the correlation between the signal and idler $\langle {\hat{a}}_{S}{\hat{a}}_{I}\rangle$, the photon numbers of signal and idler N_s_ and N_I_, gains of signal and idler G_S_ and G_I_, amplifier gain G^A^, detection gain G^D^, and other parameters mentioned in the appendix. Finally, by placing the relevant parameters according to Table 1 in it and with a straightforward calculation, the results of this simulation can be found in the following Figures shown. Additionally, the simulation results for ROC and range are obtained in the same way.

## 4. Conclusions

In this study, a proposed QTMS (quantum two-mode squeezed) radar based on an EJPA (engineered Josephson parametric amplifier) quantum source was designed and simulated, and its performance was evaluated and compared to two other radar scenarios: the conventional JPA quantum source (latest version of QTMS radar) [1] and the Josephson ring modulator (JRM) [15]. The application of an EJPA in the QTMS radar led to a significant overall improvement in radar performance. The correlation between the signal and idler was the most substantial part of QR. The greater the correlation, the better the performance of QR can be. Therefore, we must fabricate JPAs that can generate signals and idlers with much higher correlations (by engineering the JPAs). From our findings, the SNR of the QTMS radar showed a performance efficiency of about 5 dB relative to the JRM [15] and about 6 dB relative to the conventional JPA. It means that the QTMS radar, compared to CR, showed an enhancement much larger than 6 dB [2,3,4,5,6,7,8,9,15,19,20], and this study purports to have an additional 6 dB above those results [1]. The detection probability was also remarkably higher than for the other two considered scenarios. Moreover, our channel numbers in the detection probability were much lower, compared with the other two scenarios. Additionally, in this paper, we considered two detectors, A and B, which corresponded to the 5 and 1 detectors in [1,36], respectively. We compared the conventional classical radar and the QTMS radar with the EJPA design under the same conditions, showing that the QTMS radar in detector B was better than the conventional classical radar. However, in detector A, we did not achieve any quantum advantage. Hence, the choice of the detector in the QTMS radar is a significant way to improve performance. Furthermore, the QTMS radar range equation we defined showed that as the range of the QTMS radar increased, the SNR of the QTMS radar decreased proportionally. In previous works [1,3,15] for the QTMS radar, the signal was not transmitted to the target (because of its low power) [1,3], and only the transmitter and receiver antennas were facing each other, similar to the results in [15] for a low-power short-range radar (1 m due to low power). Instead, in our work, the transmission of the signal to the target was simulated over long distances. Finally, we simulated the transmission of the signal to the target and obtained a remarkable increase of 482 m for the QTMS radar range in EJPA (with signal transmission to the target), which suggests that the range can reach the order of a few hundred meters. Therefore, we have confirmed that the use of EJPA in the QTMS radar is very promising to achieve a considerable improvement in radar performance.

## Figures and Tables

**Figure 1 entropy-25-01368-f001:**
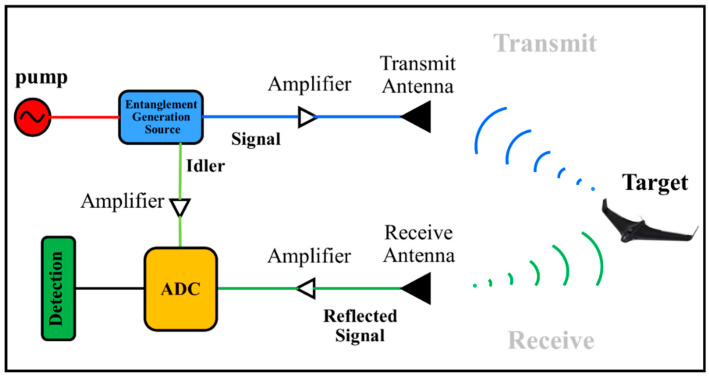
Schematic block diagram of a QR.

**Figure 2 entropy-25-01368-f002:**
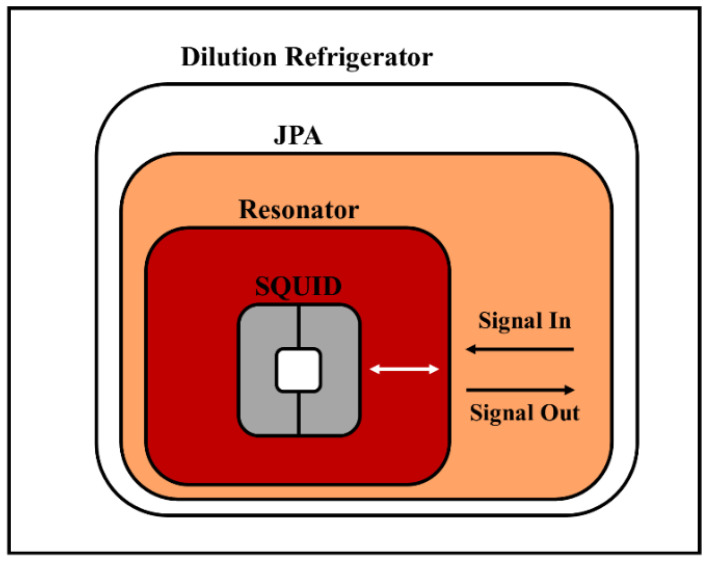
Simplified schematic representation of a JPA in a dilution refrigerator.

**Figure 3 entropy-25-01368-f003:**
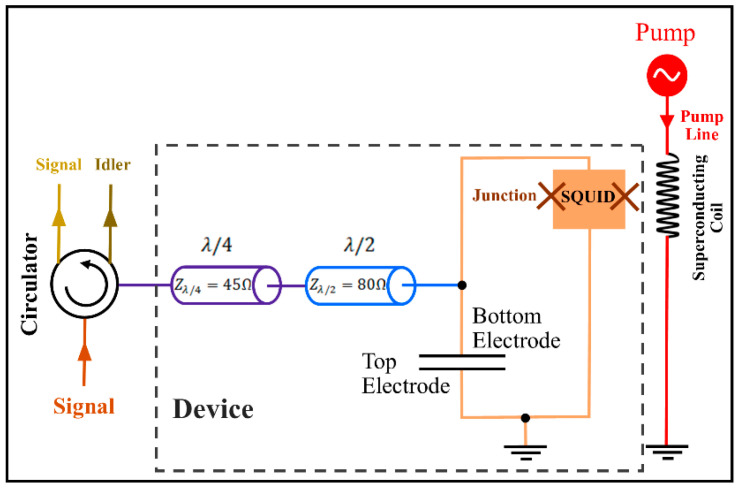
Schematic representation of an EJPA circuit.

**Figure 4 entropy-25-01368-f004:**
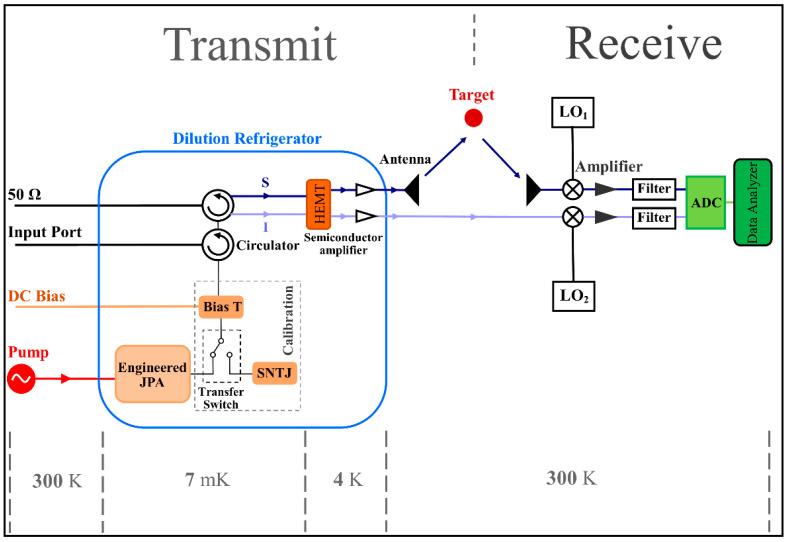
Block diagram of the QR using EJPA.

**Figure 5 entropy-25-01368-f005:**
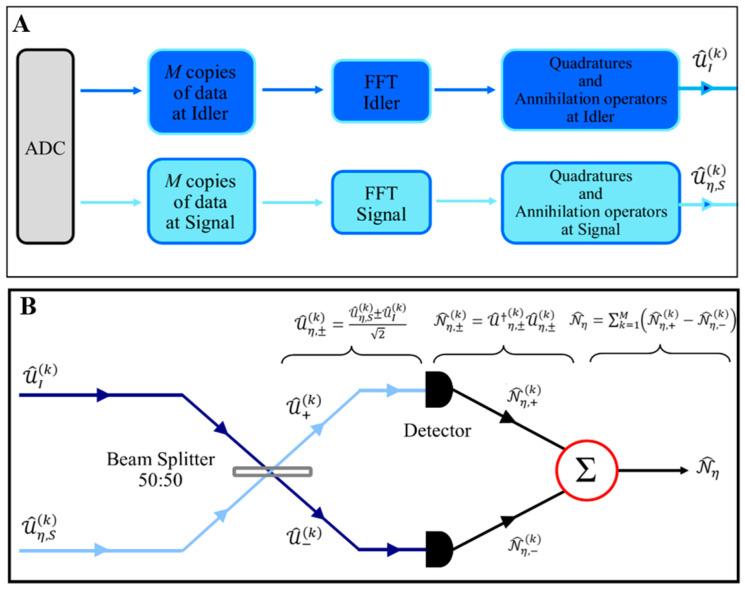
The representation of the post-processing. (**A**) The recorded data from the ADC. (**B**) Inferring the SNR of QTMS radar.

**Figure 6 entropy-25-01368-f006:**
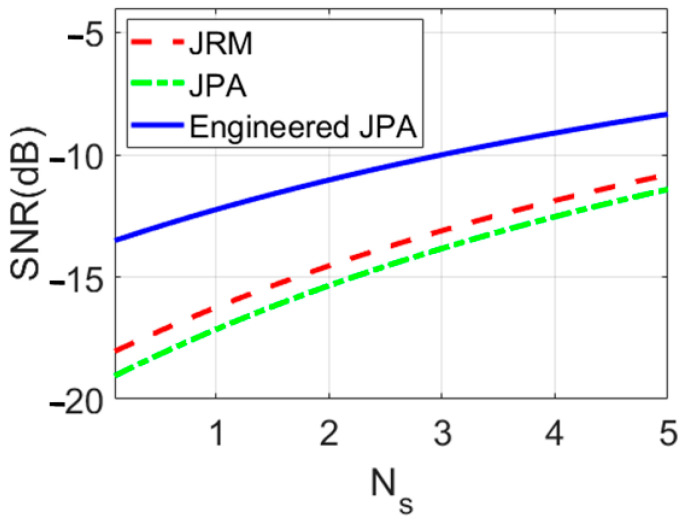
Comparison of SNRs for conventional JPA, JRM, and EJPA versus signal photon numbers N_s_.

**Figure 7 entropy-25-01368-f007:**
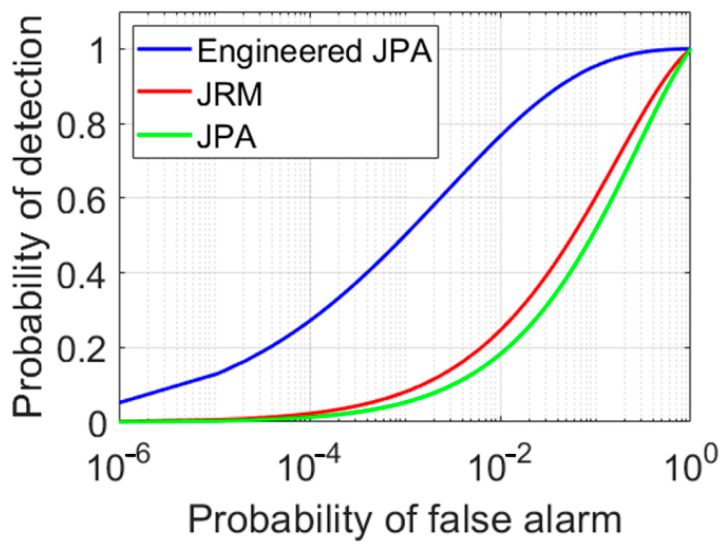
The ROC comparison plot. Comparison between conventional JPA (green), JRM (red), and EJPA (blue) for *N_s_* = 0.1, *N* = 150, and *ρ*_0_ = 1.

**Figure 8 entropy-25-01368-f008:**
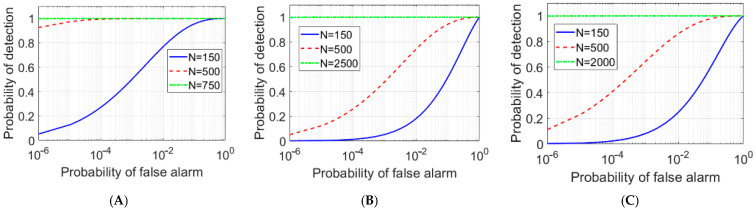
The ROC comparison plots of EJPA (**A**), conventional JPA (**B**), and JRM (**C**), with different channel numbers N.

**Figure 9 entropy-25-01368-f009:**
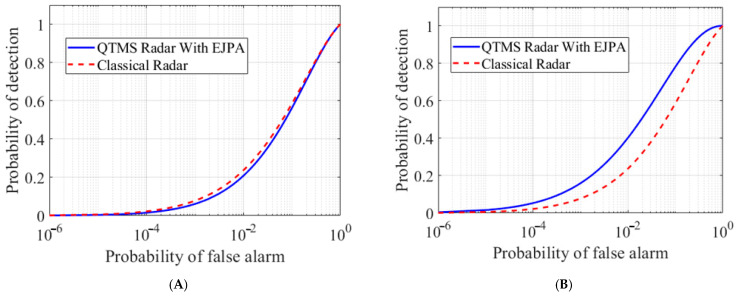
The ROC comparison plot. Comparison between the conventional classical radar (red) and the QTMS radar with EJPA (blue) with respect to detector A (**A**) and detector B (**B**), under the same condition, for *SNR*= −13.48 dB, *N* = 50, and *ρ*_0_ = 1.

**Figure 10 entropy-25-01368-f010:**
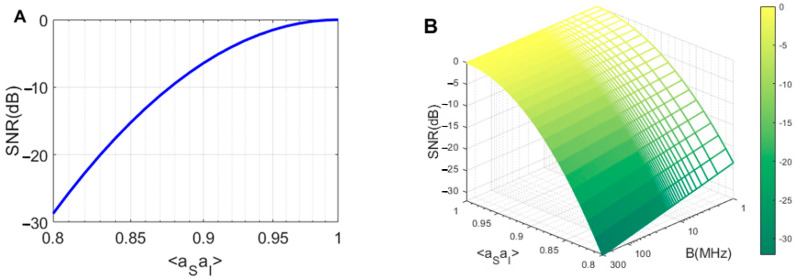
SNR plot as (**A**) a correlation function $\langle {\hat{a}}_{S}{\hat{a}}_{I}\rangle$ for an EJPA and (**B**) the correlation function and bandwidth B (MHz).

**Figure 11 entropy-25-01368-f011:**
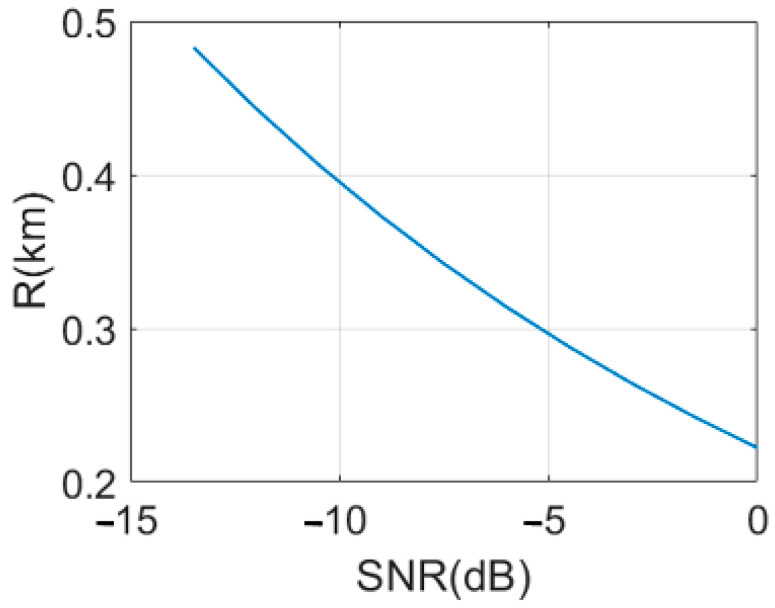
Detection range versus SNR of EJPA.

**Table 1 entropy-25-01368-t001:** The calculated parameters of QTMS radar. * Calculated parameters based on the results reported in [15]. ^†^ Calculated parameters based on the results reported in [1]. ^‡^ Parameters used from [22].

Quantity	EJPA [This Work]	JRM [15]	JPA [1]	Unit
Antenna	C-band	X-band	X-band	---
Antenna gain (G)	6.4	15 *	15	dB
Antenna effective area (A_e_)	8.8 × 10^−5^	-	-	m^2^
Target radar cross-section (*σ*)	1.0	-	-	m^2^
Bandwidth (*B*)	300 ^‡^	20	1.0	MHz
JPA or JPC power gain (G_p_)	20	30	20 ^†^	dB
HEMT gain (at 4 K) (G_HEMT_)	38	36	36 ^†^	dB
Signal gain (*G_S_*)	83.98	93.98	~96 ^†^	dB
Detection gain (*G^D^*)	16.82	16.82	16.82 ^†^	dB
Amplifier gain (*G^A^*)	67.16	77.16	79.18 ^†^	dB
Signal power (*P_s_*)	5 ^‡^	−128 *	−82	dBm
Pump power (*P_p_*)	6 ^‡^	−97	-	dBm
Noise power (*P_n_*)	−145	4	−94	dBm
Pump frequency ω_p_ = ω_s_ + ω_i_	10.62	16.89	13.6821 ^†^	GHz
Signal frequency (ω_s_)	5.31	10.09	7.5376	GHz
Idler frequency (ω_i_)	5.31	6.8	6.1445	GHz
Signal-to-noise ratio (SNR^QR^)	−13.48	−18	−19	dB
Range (R) (N_s_ = 0.1, *η* = 1 dB)	482 with signal transmission to target	1 with signal transmission to target (a low-power short-range radar)	0.5 without signal transmission to target (a low-power prototype radar)	m

## Data Availability

The data presented in this study are available within the article.

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
