# Peer review of "Investigation of the JPA-Bandwidth Improvement in the Performance of the QTMS Radar"

_entropy, 2023, doi:10.3390/e25101368_

Round 1

Reviewer 1 Report

The paper studies radar detection using the so-called quantum two-mode squeezed radar (QTMS), which detects the presence of a target by means of the joint detection of two entangled beams, generated by a Josephson parametric amplifier (JPA).  

The paper is not clearly written, it combines various elements from different approaches, and it is difficult to understand which are the novel and original aspects of this theory proposal. I cannot recommend publication of the paper in the present form, and the  manuscript needs a significant revision. If the new version is more clear and the novel aspects are better identified, I could reconsider it for publication. Let us now study in detail the various weaknesses.

1. The introduction suggests that the main original result is the evaluation of the range of the QMTS when combined with the numbers obtained from the large bandwidth JPA of Cleland group of Ref. 22.  If this is so, it does not seem to me a particularly novel result since the authors apply the approach of ref 32, 33, 35, 36 by the Balaji group and simply use the larger bandwidth demonstrated in Ref. 22. 

Actually in the following sections the authors provide much more details on the overall scheme to use, and one cannot exclude that there are other original results, but it is hard to get it because the author mix different notations and approaches. In fact, in section 3.2 and 3.3 they propose a specific detection design which coincides with the digital phase-conjugate receiver of Ref. 15, which is very different from the one used in Ref. 32, 33, 35 and 36, instead used by the authors for estimating the range. In this latter case in fact, one has simply to reconstruct the covariance matrix of the signal and idler beams, and the phase conjugate receiver is not needed. The final result of this approach is that it is not clear to me which is the specific detection scheme (if any) the authors consider and suggest. 

Moreover, there are some wrong statements in the paper which increase the confusion. For example in lines 170-173 they say:

"Note that in this work, we don't have a delay line here, but there is a time delay to measure the signal and idler [1]. The two pulses are measured at different times (does not require joint measurement). The time delay between the two pulses is due to the length of the free-space path of the transmitted signal [1]."

This is wrong. If one has to exploit the signal-idler entanglement of the pulses, they must be measured simultaneously, or at least recorded taking into account the proper delay, otherwise we are trying to look at correlations between different temporal modes of the generated microwave fields, which are uncorrelated. I cannot exclude that this comes from an erroneus understanding of the digital post-processing of ref. 15. In that paper the phase conjugated receiver is SIMULATED because after calibration and noise subtraction on heterodyne data from the measurement of the return mode, the authors use these post-processed effective modes to construct the quantities of the phase conjugate receiver. 

In the text starting from Eq. (6), the situation becomes more confusing. Eq. (6) seems to refer to the approach of Ref. 15 and it is valid in the case in which one only wants to minimize the total error rate. But eq (6) seems to me useless since from Eq. (7) and subsequent ones, the authors consider the usual Neyman-Pearson approach and look for ROC curves, in the case in which the desired detected quantity is the correlation coefficient \rho. 

From Eq. (9) one already sees the obvious effect of increasing the bandwidth. It proportionally improves the SNR simply because it increases the number of interrogating correlated pulses M. This is why in my opinion simply replacing a JPA with an engineered JPA is only an immediate and obvious result for a theory proposal. 

The main results of the paper seem to be fig. 6 to fig. 10, obtained starting from the data of Table 15 (??, should be Table 1 ??) .

It is not at all clear to me how the authors calculate these curves. I suppose N_s is the mean signal and idler number of photons per pulse, but how it is related to the parameters used in the table ? N_s is never defined in the paper. More in general, it is never explained how the table parameters are used in order to derive the plots. This should be given eventually in an appendix if it is too cumbersome. This is also particularly relevant for figure 10 which introduces the phase sensitive correlation of the JPA source <a_s a_I>, but again it is not clear at all how this is related to the parameters used in the table. 

The parameter N introduced in eq (7) I suppose it coincides with M = number of modes/pulses. Is this true ? Or not ? Further missing definitions: what are "detectors A and B" in Fig. 9 ? Moreover, what is \eta = 1 dB in the last row of Table 15 ?

A minor comment is related to subsection 2.1.1. It tries to describes the quantum properties of the QTMS radar but I find this part very confusing and not helpful at all. The sentence

"The wave packet, in another word, is compressed or squeezed inside a potential well [23], or, to be more precise, quantum noise decreases in linear compounds of some of the quadrature and increases in other compounds [1-3, 5, 6, 8]." is totally out of context. It refers apparently to a single squeezed particle, and it does not help to understand the quantum correlations of a QTMS state in which two signal-idler quadrature pairs are strongly correlated in the sense that their combination is squeezed (the variance of the difference/sum tends to zero). 

Also the sentence "When the field is quantum, in an absolute vacuum, particles can fluctuate, and these fluctuations lead to spontaneous emission from an excited state at an energy level to the ground state at the energy level , thus emitting a photon with an energy equal to the difference between these two levels, [1-3, 5, 6, 8, 23]." seems to me useless in the present context. 

In conclusion, the paper needs extensive revision. I can reconsider it for publication only if the authors provide a clear, consistent, description of their calculations, also clearly describing what are the novel results with respect to the papers by the Balaji group. 

The english is sufficiently good. 

Author Response

Reply to Referee 1

  1. Referee 1: “The introduction suggests that the main original result is the evaluation of the range of the QMTS when combined with the numbers obtained from the large bandwidth JPA of Cleland group of Ref. 22. If this is so, it does not seem to me a particularly novel result since the authors apply the approach of ref 32, 33, 35, 36 by the Balaji group and simply use the larger bandwidth demonstrated in Ref. 22.” 

Reply:

As you know, the quantum radars that have been practically implemented [1,15] so far, have a range of about 1 meter, and unfortunately, there was no specific equation to find the range for them. Furthermore, based on what Balaji mentioned in his articles, this idea is still raw and needs improvement [1,2,5,6,8]. This is of no advantage to a radar engineer. Therefore, in this paper, we show that the range of the radar reaches 482 meters, which is a great improvement for quantum radar, by using JPA's performance improvement by engineering its bandwidth. We define a specific equation for quantum radar range with respect to quantum SNR. In addition, our design and simulation use C-band antennas, which compared to recent works where antennas are in X-band, has significant advantages such as less loss, which distinguishes our work from other quantum radars. In addition, these antennas have less gain than X-band antennas, which makes us have less signal gain. Another point is that we investigated the role of the detector in these radars to better understand the quantum advantage. Finally, according to the parameters calculated from Tab. 1 using Eqs. (7, 9, 10, and 11-18), we improved the range, SNR, ROC, and quantum correlation of the prototype radars designed in the works of Balaji [1,3,6] and Barzanjeh [15], which play a very important role in the design of quantum radars. Improving the bandwidth will indeed improve the quantum radar, but this is not the whole story, because this bandwidth is related to other parameters, including the signal power, and all radar parameters must be tuned simultaneously. And the last point is that it is true that large bandwidth is stated in Ref. [22], but no practical quantum radar has been designed using a JPA with a bandwidth higher than 1 MHz, but we described a quantum radar with a bandwidth of 300 MHz and high power, which is very important from an engineering point of view.

  1. Referee 1: “Actually in the following sections the authors provide much more details on the overall scheme to use, and one cannot exclude that there are other original results, but it is hard to get it because the author mix different notations and approaches. In fact, in section 3.2 and 3.3 they propose a specific detection design which coincides with the digital phase-conjugate receiver of Ref. 15, which is very different from the one used in Ref. 32, 33, 35 and 36, instead used by the authors for estimating the range. In this latter case in fact, one has simply to reconstruct the covariance matrix of the signal and idler beams, and the phase conjugate receiver is not needed. The final result of this approach is that it is not clear to me which is the specific detection scheme (if any) the authors consider and suggest.” 

Reply:

We agree with the referee that this aspect should be more clarified. As the honorable referee mentioned, there is no need for a phase-conjugate receiver here, therefore, in SubSec. 3.3, the following Figure has been added and the sentences of the first and second paragraphs have been edited as below:

A

B

Figure 5. The representation of the post-processing.

“As we shown in Fig. 5 (A and B), the idler and signal modes are recorded after amplification with an analog-to-digital converter (ADC) card (our suggestion to experimental researchers is to use dual-channel ADC AD570JD with 8-bit resolution) [15]. The recorded data from the ADC is split into shorter arrays of copies. To derive the measurement statistics of the signal and idler mode quadratures can be used the digital fast Fourier transform (FFT) at idler () and signal frequencies () after analog down conversion on each array separately. These measurement results are useful to compute the covariances of the signal and idler modes.”

“The M copies of the signal and idler modes from ADC are sent to the detectors (Fig. 5 B). ...”

  1. Referee 1: “Moreover, there are some wrong statements in the paper which increase the confusion. For example in lines 170-173 they say:

"Note that in this work, we don't have a delay line here, but there is a time delay to measure the signal and idler [1]. The two pulses are measured at different times (does not require joint measurement). The time delay between the two pulses is due to the length of the free-space path of the transmitted signal [1]."

This is wrong. If one has to exploit the signal-idler entanglement of the pulses, they must be measured simultaneously, or at least recorded taking into account the proper delay, otherwise we are trying to look at correlations between different temporal modes of the generated microwave fields, which are uncorrelated. I cannot exclude that this comes from an erroneus understanding of the digital post-processing of ref. 15. In that paper the phase conjugated receiver is SIMULATED because after calibration and noise subtraction on heterodyne data from the measurement of the return mode, the authors use these post-processed effective modes to construct the quantities of the phase conjugate receiver.”

Reply:

Since our work is very similar to Balaji's work in Ref. [1] in the topic of signal transmission and idler recordation, therefore, as mentioned in this reference, we do not use joint measurement and use time delay instead of delay line. However, the correlation between the signal and the idler does not disappear. Therefore, we disagree with this referee's comment. In the following, we draw your attention to the text of the Ref. [1]:

  1. Referee 1: “In the text starting from Eq. (6), the situation becomes more confusing. Eq. (6) seems to refer to the approach of Ref. 15 and it is valid in the case in which one only wants to minimize the total error rate. But eq (6) seems to me useless since from Eq. (7) and subsequent ones, the authors consider the usual Neyman-Pearson approach and look for ROC curves, in the case in which the desired detected quantity is the correlation coefficient \rho.

From Eq. (9) one already sees the obvious effect of increasing the bandwidth. It proportionally improves the SNR simply because it increases the number of interrogating correlated pulses M. This is why in my opinion simply replacing a JPA with an engineered JPA is only an immediate and obvious result for a theory proposal. ”  

Reply:

We agree with the referee's opinion and Eq. 6 was completely removed. In response to the second part of the reviewer's comment, it can be stated that the improvement of SNR does not happen only by increasing the bandwidth, as we can see from Eqs. (4, 5, 8 and A1-A8) and Tab.1, SNR is related to other parameters besides the bandwidth. We need to consider other parameters such as signal power, JPA gain, antenna gain, etc. at the same time as increasing the bandwidth to achieve the desired improvement. In addition to the improvement in SNR, we also have an improvement in ROC, correlation, and range, which is poorly understood from the formula. Hence, we need to simulate them to better understand the improvement as described in Figs. (7-11).

  1. Referee 1: “The main results of the paper seem to be fig. 6 to fig. 10, obtained starting from the data of Table 15 (??, should be Table 1 ??) .”

Reply: Corrected.

  1. Referee 1: “It is not at all clear to me how the authors calculate these curves. I suppose N_s is the mean signal and idler number of photons per pulse, but how it is related to the parameters used in the table ? N_s is never defined in the paper. More in general, it is never explained how the table parameters are used in order to derive the plots. This should be given eventually in an appendix if it is too cumbersome. This is also particularly relevant for figure 10 which introduces the phase sensitive correlation of the JPA source <a_s a_I>, but again it is not clear at all how this is related to the parameters used in the table.” 

Reply:

We agree with the referee that this aspect should be more clarified. Therefore, in SubSec. 3.4, the following paragraph has been added as below:

“The question raised here is how the parameters in Table 1 are extracted and how the simulation is done in this paper. It can be answered like this by considering equation 8 for example. All the required expressions are given in the appendix by Eqs. (A1-A8) so that by placing them in Eq. 8, we obtain a general relation for SNR, which includes the correlation between signal and idler, the photons number of signal and idler Ns and NI, gains of signal and idler GS and GI, amplifier gain GA, detection gain GD and other parameters mentioned in the appendix. Finally, by placing the relevant parameters according to Table 1 in it and a straightforward calculation, the results of this simulation can be found in the following Figures shown. Also, the simulation results for ROC and range are obtained in the same way.”

  1. Referee 1: “The parameter N introduced in eq (7) I suppose it coincides with M = number of modes/pulses. Is this true ? Or not ? Further missing definitions: what are "detectors A and B" in Fig. 9 ? Moreover, what is \eta = 1 dB in the last row of Table 15 ?”

Reply:

N is the number of samples integrated and defined below of Eq. 6. The detector A is the same as Eq. 6 (in revised manuscript) to which we added a subscript A for clarity. Also, detector B is described at the top of Fig. 9 and we added a subscript B to it. Moreover, note that the η defined below of Eq. 5 such that it is effective dissipation range. Thanks for your notice.

  1. Referee 1: “A minor comment is related to subsection 2.1.1. It tries to describes the quantum properties of the QTMS radar but I find this part very confusing and not helpful at all. The sentence

"The wave packet, in another word, is compressed or squeezed inside a potential well [23], or, to be more precise, quantum noise decreases in linear compounds of some of the quadrature and increases in other compounds [1-3, 5, 6, 8]." is totally out of context. It refers apparently to a single squeezed particle, and it does not help to understand the quantum correlations of a QTMS state in which two signal-idler quadrature pairs are strongly correlated in the sense that their combination is squeezed (the variance of the difference/sum tends to zero). 

Also the sentence "When the field is quantum, in an absolute vacuum, particles can fluctuate, and these fluctuations lead to spontaneous emission from an excited state at an energy level to the ground state at the energy level , thus emitting a photon with an energy equal to the difference between these two levels, [1-3, 5, 6, 8, 23]." seems to me useless in the present context.” 

Reply:

We agree with the referee's opinion and have removed these sentences completely. Thanks for your notice.

  1. Referee 1: “In conclusion, the paper needs extensive revision. I can reconsider it for publication only if the authors provide a clear, consistent, description of their calculations, also clearly describing what are the novel results with respect to the papers by the Balaji group.” 

Reply:

The difference between our paper and the works of Balaji [1,3] and Barzanjeh [15] is that we used an engineered JPA in our quantum radar design, which has both high power and high bandwidth, in addition, we used a C-band antenna. The other thing is that we send the signal to the target, in case there is no target at all in Balaji and Barzanjeh's works, to which they can send the signal, and only the receiving antenna is placed in front of the transmitting antenna and the signal is sent within half a meter to one meter and have received the transmitted signal. In such a situation, when we send a signal to the target, we revealed a significant improvement in the performance of the quantum radar compared to other existing quantum radars. The range of the quantum radar reached 482 meters (according to the range equation we defined), the SNR was improved by several dB, the ROC was greatly improved, and we also determined what the role of bandwidth is in the correlation. We are glad that the respected reviewer is familiar with the literature on quantum radars so that it is easy for (him/her) to understand such an improvement. These results are very important in telecommunication engineering. Of course, there is a long way to such improvements, and our article can be a motivation for improving the performance of quantum radars with high power and long operating range. Note that all calculations are theoretically achievable and all data generated or analyzed during this study are included in this paper. We tried to improve all aspects of our work corresponding to referee comments and edited the manuscript wherever needed. In addition, you can see the answers to the 1 and 6 comments. In addition, a recent article from Nature Physics, has benefited from our results, provided in a preprint of our manuscript: (Assouly, R., Dassonneville, R., Peronnin, T., Bienfait, A. and Huard, B., 2023. Quantum advantage in microwave quantum radar. Nature Physics, pp.1-5.).

Thanks for your attention.

Author Response

Reply to Referee #2

  1. Referee 2: “The authors claim to conduct a simulation of the enhanced performances of the quantum radar using the EJPA. However, there is no section providing a concise theoretical background or explaining the simulation methodology. Equations (1)-(9) are presented without any contextual information, making it difficult to understand what is being simulated and how it is executed.”
  2. Referee 2: “Throughout section 3, the manuscript lists a series of technical features for a hypothetical setup of EJPA. However, it remains unclear whether this experimental setup is realized in practice and how it is used for simulating the improved radar. The introduction mentions that ”we present and evaluate the EJPA and use it to simulate the radar’s design,” but it is unclear what the evaluation of EJPA entails and how it is connected to the simulation. I kindly request the authors to revise the manuscript, providing a clear separation into three distinct sections. The first section should present the minimal theoretical background, providing essential context for the readers. The second section should focus on listing the technical features of the EPJA and explaining how it is utilized in the simulation process. Lastly, the third section should comprehensively present the results obtained from the simulations. This organization will greatly enhance the manuscript’s accessibility and make it easier for readers to follow and understand the research.”

Reply to comment 1 and 2:

We agree with the referee that this aspect should be more clarified. Therefore, in SubSecs. 3.2 and 3.4, the following paragraphs have been added as below:

“The main idea of using EJPA comes from the fact that we need a high-range QR in practice. Our EJPA has three wings that are very useful for improving our simulated quantum radar: first, high bandwidth, second, high power, and third, low gain. In the design of a QR, special attention should be paid to very important points, including various parameters of the radar. For example, to implement a QR with a long range, we need high signal power, of course, how high the power should be so as not to suppress the correlation. Or what antennas should we use with what gains or what amplifiers should we use that will introduce less noise into the system? Therefore, all the parameters of a QR must match each other to find an improvement in its performance. In this paper, we investigate the improvement of a QR according to the sensitivity of choosing different parameters.”

“The question raised here is how the parameters in Table 1 are extracted and how the simulation is done in this paper. It can be answered like this by considering equation 8 for example. All the required expressions are given in the appendix by Eqs. (A1-A8) so that by placing them in Eq. 8, we obtain a general relation for SNR, which includes the correlation between signal and idler, the photons number of signal and idler Ns and NI, gains of signal and idler GS and GI, amplifier gain GA, detection gain GD and other parameters mentioned in the appendix. Finally, by placing the relevant parameters according to Table 1 in it and a straightforward calculation, the results of this simulation can be found in the following Figures shown. Also, the simulation results for ROC and range are obtained in the same way.”

  1. Referee 2: “It is advisable to avoid using very generic terms, such as ”quantum theory” in the abstract (third line), or providing a long list of phenomena (e.g., ”the principle of uncertainty, entanglement, correlation, photon statistics, vacuum fluctuations, and squeezing are present in QRs”) in the introduction. Instead, a more focused approach would be preferable, providing a concise survey of how specific phenomena actually improve the performances of classical radars.”

Reply:

According to the referee's comment, the sentences were corrected. In addition, for the second part of the comment, the answer can be stated as follows, since Balaji et al. [1-3,5-9,32,33,35,36] have already examined the differences between quantum and classical radars, in this work, we avoid presenting those results again and only we present the comparison between the quantum radars which has been implemented so far and the current plan. Thanks for your attention.

  1. Referee 2: “In section 2.1.1, the following sentence is unclear and imprecise: ”The term vacuum here refers to vacuum or zero-point energy-related fluctuations, which we intend to amplify. These fluctuations are not similar to classical ones [1, 3]. When the field is quantum, in an absolute 1vacuum, particles can fluctuate, and these fluctuations lead to spontaneous emission from an excited state at an energy level to the ground state at the energy level, thus emitting a photon with an energy equal to the difference between these two levels [1-3, 5, 6, 8, 23].”I recommend rephrasing it to enhance clarity and accuracy.”

Reply:

We agree with the referee's opinion and have removed this sentence completely. Thanks for your notice.

  1. Referee 2: “In section 2.2, the sentence “resulting in the squeezing of conjugate signals I and Q” requires improvement. First, provide a more precise description of squeezing, emphasizing that it is not solely a consequence of entanglement. Second, introduce the notation ”I and Q” with prior explanation to ensure readers can understand its meaning.”

Reply:

We agree with the referee that this aspect should be more clarified. Therefore, we edited the sentence in SubSec. 2.2 and the following explanations have been added in the second paragraph in SubSubSec. 2.1.1 and SubSec. 2.2:

In this paper, we deal the correlating and squeezing in-phase (I) and quadrature (Q) voltages. We can brief the operation of the QTMS radar as follows:

  1. Utilize the JPA to generate a pair entangled signal (Signal and Idler). Amplify both signal and idler. Transmit signal through free space forward to the target. Do a heterodyne measurement on the idler and hold a record of the results in the form of time series of I and Q voltages.
  2. Receive a reflected signal. Fulfill a heterodyne measurement on it to create time series of I and Q voltages.
  3. Correlate the I and Q voltages of the signal and idler.
  4. If the correlation surpasses a preset threshold, notify a detection.”

“As mentioned in 2.1.1, since the photons of the signal and idler originate from the same pump photon, there is a strong quantum correlation between the signal and idler, resulting in the squeezing of I and Q voltages [1-3, 5, 6, 8, 15]. Important to emphasize that, the squeezing is not solely a consequence of entanglement.”

  1. Referee 2: “Other imprecise sentences are present all along the manuscript. I ask the authors to check again and improve the manuscript where is needed.”

Reply:

We thank the referee for (him/her) significant comments and state that we have made a major revision throughout the manuscript.

Round 2

Reviewer 1 Report

The new version has significantly improved the clarity of presentation and most of the issues I have raised in my first report have been properly addressed by the authors. As a consequence I think that the paper can be published in Entropy, even though I think there is still room for further clarity and improvements.

More in detail:   

1. The paper now more clearly and more properly defines the calculations of the paper as simulation of the performance of the proposed quantum radar. However, it must be more explicitly stated that a range of 482 meters is just an estimation based on a set of a large number of estimated parameters, putting together devices and apparatuses of different kind. I would better say that the range can reach order of few hundred of meters, given that the error in the estimate can be well in the order of hundred of meters. 

2. The authors did not accept  my argument related to the question of joint measurements. Probably I have not been clear enough. I think that the lines 173-176 of the new version, saying "Note that in this work, we don't have a delay line here, but there is a time delay to measure the signal and idler [1]. The two pulses are measured at different times (does not require joint measurement). The time delay between the two pulses is due to the length of the free-space path of the transmitted signal [1]." 

is misleading. It is true that the assumed detector performs independent heterodyne measurements on signal and idler, but the detector exploits the QUANTUM CORRELATIONS between them and therefore it must be stressed and clarified that the correlations are taken between the properly correlated pulses, i.e. the correlated pairs generated by the EJPA and that therefore the accumulated delay must be known and properly taken into account when constructing the correlations between the idler and the correlated return mode. This is not said, and the paper gives the incorrect impression that one does not have to worry about the time delay.  This is instead crucial because if in a real experiment one constructs correlations with uncorrelated return and idler pulses will see nothing. 

In my opinion measuring the correlation between the measured quadratures of k-th correlated pair is in fact a form of joint measurement of the signal-idler pairs. 

3. The presentation of the detection scheme is much more clear but there is still confusion. Figure 5 is exactly the same PCR scheme of Ref. 15, which is a good way of measuring the correlation <a_s a_I>, and which is related to detector 1 of ref. 1.   

So in my view the scheme proposed in this paper considers the simulated PCR of ref 15 and should use only detector 1 of ref. 1. I do not understand why (and I think it is even misleading) the authors consider Detector 5 of Ref 1 (what they call detectors A) in eq 6 and in some of the plots. The PCR design of Fig. 5 does not seem good to me for such a detector but it is designed just for detector 1 of ref 1 (the one named detector B by the authors) which is related to the relevant correlation created by the EJPA, <a_s a_I>.   

This is confirmed in fact by Fig. 9. 

So in conclusion I suggest to consider only detector B and I do not see any advantage in considering detector A (and this was one the things which was confusing in the previous version in my opinion). 

A final suggestion is that I am not sure that Fig 10 is useful. Part A is quite obvious. Part B is a bit confusing in my opinion because in a real experiment the correlation <a_s a_I> and the bandwidth are input parameters of the given Josephson source. They are in general independent, or more properly, depend upon on the design and fabrication details of the Josephson circuit used.  The dependence shown here is due to having fixed other parameters (which ones ?) and  it seems ot me an improper way of showing things. 

Once these points have been clarified the paper can be published in Entropy.

only minor adjustements are needed. 

Author Response

Reply to Referees - entropy- 2500270

Dear Editor,

Please find here the revised version of our manuscript entropy-2500270 entitled: “Investigation on the JPA-bandwidth improvement in the performance of the QTMS radar”.

We really appreciate the referees for pointing out invaluable comments that led to further clarifying various aspects of our results. We have made a considerable effort to comply with their constructive suggestions, improving the formulation of our manuscript under various aspects.

In the following, we list our replies to the points raised by the referees and the corresponding changes made in the manuscript.

------------------------------------------------------------------------------

Reply to Referee 1

  1. Referee 1: “The paper now more clearly and more properly defines the calculations of the paper as simulation of the performance of the proposed quantum radar. However, it must be more explicitly stated that a range of 482 meters is just an estimation based on a set of a large number of estimated parameters, putting together devices and apparatuses of different kind. I would better say that the range can reach order of few hundred of meters, given that the error in the estimate can be well in the order of hundred of meters.” 

Reply:

We agree with the referee that this aspect should be clarified. Therefore, in SubSec. 3.4, the following paragraph has been added as below:

“Note that a range of 482 meters is just an estimation based on a set of a large number of estimated parameters, putting together devices and apparatuses of different kinds. In other words, the range can reach the order of a few hundred meters.”

In addition, in the conclusion, we add a related sentence to clarify.

  1. Referee 1: “The authors did not accept my argument related to the question of joint measurements. Probably I have not been clear enough. I think that the lines 173-176 of the new version, saying "Note that in this work, we don't have a delay line here, but there is a time delay to measure the signal and idler [1]. The two pulses are measured at different times (does not require joint measurement). The time delay between the two pulses is due to the length of the free-space path of the transmitted signal [1]."

is misleading. It is true that the assumed detector performs independent heterodyne measurements on signal and idler, but the detector exploits the QUANTUM CORRELATIONS between them and therefore it must be stressed and clarified that the correlations are taken between the properly correlated pulses, i.e. the correlated pairs generated by the EJPA and that therefore the accumulated delay must be known and properly taken into account when constructing the correlations between the idler and the correlated return mode. This is not said, and the paper gives the incorrect impression that one does not have to worry about the time delay.  This is instead crucial because if in a real experiment one constructs correlations with uncorrelated return and idler pulses will see nothing.

In my opinion measuring the correlation between the measured quadratures of k-th correlated pair is in fact a form of joint measurement of the signal-idler pairs.” 

Reply:

We agree with the referee that this aspect should be clarified. Therefore, we edited the paragraph in question according to the referee's useful comment as follows:

“Note that in this work, we don't have a delay line here, but there is a time delay to measure the signal and idler [1]. The two pulses are measured at different times with respect to maintaining the correlation between them. The time delay between the two pulses is due to the length of the free-space path of the transmitted signal [1].”

  1. Referee 1: “The presentation of the detection scheme is much more clear but there is still confusion. Figure 5 is exactly the same PCR scheme of Ref. 15, which is a good way of measuring the correlation <a_s a_I>, and which is related to detector 1 of Ref. 1.

So in my view the scheme proposed in this paper considers the simulated PCR of ref 15 and should use only detector 1 of ref. 1. I do not understand why (and I think it is even misleading) the authors consider Detector 5 of Ref 1 (what they call detectors A) in eq 6 and in some of the plots. The PCR design of Fig. 5 does not seem good to me for such a detector but it is designed just for detector 1 of ref 1 (the one named detector B by the authors) which is related to the relevant correlation created by the EJPA, <a_s a_I>.  

This is confirmed in fact by Fig. 9.

So in conclusion I suggest to consider only detector B and I do not see any advantage in considering detector A (and this was one the things which was confusing in the previous version in my opinion).”

Reply:

This referee's comment is very helpful. As you know, there is no limit to the use of various detection functions as mentioned in Ref. [1]. In this article, detector A was used because Refs. [1] and [15] used this detector to check the probability of error and we also compared the improvement of EJPA using this detector with Refs. [1] and [15]. These results state that the ROC plots in EJPA with specific detector A have a better advantage than JRM and conventional JPA with the same specific detector A. However, it does not have a better advantage compared to the classical detector. So, it was necessary to introduce another detector that would show a quantum advantage over its classical counterpart that detector B could easily be used. In addition, we intended to explain the very important result of "how to choose the correct detector" well. Therefore, a very important result that can be used in practical work is that detectors have different functions in different conditions and they should be chosen correctly according to the needs.

      So, in summary, EJPA with detector A has a better advantage than the two scenarios of conventional JPA and JRM in Refs. [1] and [15], respectively. However, it was observed that it is not superior to its classical counterpart. But detector B has a significant advantage over its classical counterpart. These results are very important in practical work. Therefore, Fig. 9 contains significant results and needs to be included in the article. Thank you for your attention

  1. Referee 1: “A final suggestion is that I am not sure that Fig 10 is useful. Part A is quite obvious. Part B is a bit confusing in my opinion because in a real experiment the correlation <a_s a_I> and the bandwidth are input parameters of the given Josephson source. They are in general independent, or more properly, depend upon on the design and fabrication details of the Josephson circuit used. The dependence shown here is due to having fixed other parameters (which ones ?) and it seems to me an improper way of showing things. ”  

Reply:

We agree with the referee's comment to clarify this subject and have shown Fig. 10 B in a clearer way. As you know, SNR in Eqs. A3 and A4 in the appendix, it is evident that it depends on the correlation, and in Eq. 8, it depends on the bandwidth. This correlation is related to the correlation between the signal and the idler in the SNR that is at the input of the detector. Therefore, we have illustrated the 3D plot of this dependence in Fig. 10 B, which can be used to improve the performance of quantum radars in practical work. This figure shows that JPA engineering is not only related to bandwidth; other important factors such as maintaining correlation are also very important in radar SNR. Therefore, these results are very useful in the design of operational radars. Hence, we presented both Fig. 10 B and its results more clearly in the new version. Thank you for your timely notice. 

Author Response

Reply to Referee #2

  1. Referee 2: “explicitly asked to insert some theoretical background in a dedicated section in such a way the reader can appreciate the relevance of the experimental setup and the meaning of the simulation. It is not possible to consider equation (8) or the bunch of equations in the appendix as an answer to this objection. The equations (1)-(8) are presented without an appropriate context.

     Moreover, the authors keep on mixing terminology, suggesting the use of an explicit experimental setup (”In this study, we used C band antennas (4-8 GHz)” page 9 just to mention one case), which is in tension with the fact that they just provide a simulation of the EJPA. All the lengthy experimental details provided in section 3, before any theoretical background about the simulation is provided, distract the reader from the final objective of the manuscript.”

Reply to comment 1:

Thank you for your ameliorative comment. It should be noted that the calculations were performed according to the standard approach presented in Refs. [15,34], require the description of the post-processing along with the diagram shown in Fig. 5. This means that the equations extracted from Sec. 3.3 and the appendix, make sense with these descriptions. Therefore, the paper has explained the background theory of this problem according to the standard approach. However, according to the request of the honorable referee, we moved Table 1 to the last part of the article in order to better represent the results.

     In response to the second part of the comment, it can be answered that this work is completely a theoretical study. However, due to the reviewer's helpful comments, we have edited sentences throughout the paper that were ambiguous and falsely suggestive of an experiment, so that the reader fully understands that this work is purely theoretical.

Finally, we thank the referee for valuable comments.

-----------------------------------------------------------------------------------------

In conclusion, we believe that these major changes and modifications comply with the referees' points and significantly improve the overall presentation. We also reckon that the manuscript may be now suitable for publication.

Thank you very much for your precious attention and cooperation.

Yours sincerely,

Jamlileh Seyed-Yazdi

On behalf of the Authors

Milad Norouzi,

Jamileh Seyed-Yazdi*,

Seyed Mohammad Hosseiny,

Patrizia Livreri

Round 3

Reviewer 2 Report

The authors slightly improved the discussion, that is now more clear. Thanks to the adjustments, the reader can move through the manuscript appreciating  the actual contributions of the authors. For this reason, I approve the manuscript for publication.